# Greek Landrace Flours Characteristics and Quality of Dough and Bread

**DOI:** 10.3390/foods12081618

**Published:** 2023-04-11

**Authors:** Adriana Skendi, Maria Papageorgiou, Maria Irakli, Stefanos Stefanou

**Affiliations:** 1Department of Food Science and Technology, International Hellenic University, POB 141, GR-57400 Thessaloniki, Greece; andrianaskendi@hotmail.com; 2Institute of Plant Breeding and Genetic Resources, Hellenic Agricultural Organization—Dimitra, Thermi, GR-57001 Thessaloniki, Greece; irakli@ipgrb.gr; 3Department of Agriculture, International Hellenic University, POB 141, GR-57400 Thessaloniki, Greece; stefst2@ihu.gr

**Keywords:** antioxidant activity, bread quality, dough quality, gluten, landrace, microelements, organic farming, phenolics, wheat

## Abstract

Besides organic growing, ancient wheats and landraces are attracting the attention of scientists who are reassessing the healthy and dietary properties attributed to them by popular tradition. A total of eleven wheat flours and whole meal samples were analyzed, of which, nine originated from the organic farming of five Greek landraces (one einkorn, one emmer, two durum, and one soft wheat) and a commercial organically grown emmer cultivar. Two commercial conventional flours of 70% and 100% extraction rate were examined for comparison purposes. Chemical composition, micronutrients, phenolic profile, and quantification, and antioxidant activity of all samples were determined. Moreover, dough rheology and breadmaking quality were studied; Flours from local landraces were higher in micronutrients, phenolic content, and antioxidant activity than the commercial samples. The 90% extraction flour of the landrace, besides the highest protein content (16.62%), exhibited the highest content of phenolic acids (19.14 μg/g of flour), whereas the commercial refined emmer flour was the lowest (5.92 μg/g of flour). The same milling of the einkorn landrace also showed a higher specific volume (1.9 mL/g vs. 1.7 mL/g) and lower bread crumb firmness than the whole meal commercial emmer sample (33.0 N vs. 44.9 N). The results of this study showed that the examined Greek wheat landraces could be considered as a possible source of microelements, phenolics, and antioxidants with a beneficial effect in human health, and by using an appropriate breadmaking procedure, they could produce high-quality breads.

## 1. Introduction

Emmer and einkorn are types of hulled wheats considered among the most ancient Triticeae cultivated in the world. They have been cultivated for a long time, representing a staple food for the general population, but were neglected with time. Hull represents a problem for ancient wheats, such as einkorn and emmer, when compared to bread wheat because of their low yield and extra effort to remove the hulls [1]. In recent years, there is observed a trend toward consuming more natural foods that derive from sustainable agriculture. This trend led to the rediscovery of ancient foods and flavors, and a renewed interest in these neglected for many years species and landraces. Wheat landraces with exceptional phenotypes are seen as a valuable genetic source to improve modern wheat varieties [2].

In addition to organically produced cereals, ancient wheats and landraces have been reintroduced into the bakery during the last decades because of the growing awareness of consumers about the health and organoleptic characteristics of the derived foods. In general, proline residues of gluten are the less digested from the enzymes present in the human gastrointestinal tract and some of these protein fractions considered “as harmful gluten proteins” are not present in durum wheat, emmer, and einkorn [3,4], making their derived products more easily digested from individuals showing low coeliac toxicity. Moreover, the ancient wheats are considered less allergenic than the recent common wheat [4,5].

Breeding is oriented toward high agronomic yield rather than the nutritional quality [6], while ancient wheats and landraces are considered rich in different nutrients [5,7]. In addition to being disease resistant, ancient wheats and landraces are considered a source of phytochemicals, such as phenolics, alkylresorcinols, vitamin E, sterols, etc. In their work, Ziegler et al. [8] reported that the amount of lutein, a carotenoid, among wheat species decreases in the following order: einkorn > durum > spelt > emmer > bread wheat. In another study, higher levels of vitamin A and B complex and minerals were observed in emmer compared to einkorn lines and durum wheat cultivars [9].

In breadmaking, both the amount and the quality of the protein present in the wheat flour are of great importance. Compared to common wheat, ancient wheats are associated with poor baking quality due to the poor gluten quality [10,11,12]. Intensive breeding has resulted in an increase in gluten proteins due to their importance in baking and processing [6,13]. The protein content in the old varieties is higher than in modern ones [11,12], but of lower technological quality for baking [12]. In their study Frakolaki et al. [14] reported that spelt flour has higher protein content but less gluten than wheat flour. Wheat gluten fractions considered responsible for the final volume of the bread are gliadins, glutenins, and glutenin macropolymer, which are a high molecular weight subfraction of glutenin [15]. HMW-GS (High-molecular-weight glutenin subunits) were considered the most essential determinant of bread-making quality [16] and used to indicate the genetic potential of bread wheat varieties since it affects the viscoelastic properties of dough and the baking quality [17]. The gluten from ancient wheats is reported to have lower elasticity and stability than bread wheat cultivars resulting, after kneading, in a soft and sticky dough that is difficult to handle [18]. Ancient wheats and landraces can be utilized to produce nutrient denser bakery products compared to those prepared with modern wheat varieties, but also with differences in the quality of the final product mainly due to protein quality.

Conventional modern agriculture shows preference to wheat varieties that have high yields, leading to the loss of ancient wheats (einkorn, emmer) and wheat landraces of higher nutritional value and richer genetic diversity and resistance. Thus, cultivation of these neglected wheats is an option in order to ensure the future of modern crops.

Although there exist many Greek ancient wheats and landraces, they are mostly grown on a small scale and only for in farm cultivation. So far, no reported study evaluates Greek wheat landraces based on their dough and breadmaking capacity, phenolic profile, and antioxidant activity. To meet consumers’ demand for healthy cereals and cereal products, the industry has placed in the market flours from commercial ancient wheat varieties grown according to conventional farming principles. Evaluation of these commercial varieties and comparison with local landraces could give insights into the quality of commercialized ancient wheats. This information could be handy to the baking industry assessing the possibilities that local landraces could represent.

Thus, the aim of the present study was the evaluation of the chemical composition, dough rheological behavior and baking properties of nine wheat millings from Greek landraces grown under organic farming and their comparison against two commercial emmer millings. To reach this aim we evaluated the differences in chemical composition, phenolic profile, antioxidant activity and technological quality between two millings of a bread wheat landrace *Triticum aestivum* L. (whole meal and refined flour), two *Triticum durum* landraces (whole meal and refined), one emmer landrace *Triticum dicoccum* (whole meal and refined), and one einkorn landrace *Triticum monococcum* (90% extraction rate)*,* all of the above grown under organic farming and checked against one commercial emmer sample *Triticum dicoccum* (whole meal and refined) grown under conventional farming.

## 2. Materials and Methods

### 2.1. Samples and Chemicals

A total of nine flour types (one local einkorn landrace *Triticum monococcum* L. of 90% extraction yield (M90); one emmer landrace *Triticum dicoccum*, whole meal (D100), and of 70% extraction yield (D70); two durum landraces *Triticum durum*, whole meal (DuA100) and (DuB100), and of 70% extraction yield (DuA70) and (DuB70); one local soft wheat landrace *Triticum aestivum* L., whole meal (S100), and of 70% extraction yield (S70)), were provided by Antonopoulos Farm, Larisa, Greece. The flours were obtained from the millstone milling procedure. Furthermore, two commercial emmer flours, one whole meal (DC100), and the other of 70% extraction yield (DC70) were obtained from the local market.

Methanol and water of HPLC grade, Folin–Ciocalteu reagent, sodium acetate trihydrate, and sodium hydroxide were obtained from Chem-Lab NV, (Zedelgem, Belgium). The single element standard solutions 1000 mg/L, of As, Cd, Cr, Cu, Fe, Mn, Ni, Pb, Zn, K, Na, Ca, and Mg were of Sigma Chem (St. Louis, MO, USA).

The DPPH (2,2-diphenyl-1-picrylhydrazyl) and (+)-catechin were from Sigma Aldrich (St. Louis, MO, USA). ABTS (2,2′-azinobis (3-ethylbenzothiazoline-6-sulfonic acid), Trolox ((S)-(-)-6-hydroxy-2,5,7,8- tetramethylchroman-2-carboxylic acid), and gallic acid were from J&K Scientific GmbH (Pforzheim, Germany). TPTZ (2,4,6-tripyridyl-s-triazine) and aluminumchloride-6-hydrate were from Alfa Aesar, GmbH & GoKG (Karlsruhe, Germany). Iron (III) chloride hexahydrate, sodium carbonate, and sodium nitrite was from Merck, KGaA (Darmstadt, Germany).

Standards, such as p-coumaric acid (pCA), sinapic acid (SA), caffeic acid (CA), ferulic acid (FA), and 4-hydroxybenzoicacid (4-HBA), were purchased from Sigma, (Steinheim, Germany); (+)-catechin (CAT), was purchased from Extrasynthese (Genay Cedex, France); protocatechuic acid (PRCA), was from Alfa Aesar, (Heysham, UK), whereas vanillic acid (VA) was from Fluka (Steinheim, Germany).

The freeze-dried yeast ‘‘Giotis” was a commercial brand from Giotis (Peristeri, Greece). All other reagents and chemicals used were of analytical reagent grade.

### 2.2. Flour Quality Analysis

Protein content was determined by the ICC 105/2 method [19], whereas ash content in wheat flours was measured according to the ICC 104/1 [20]. Wet and dry gluten were determined following ΙCC standard method155 [21], while falling number (FN) was determined by the approved method AACC 56–81.04 [22]. The moisture content of the flours was determined by oven drying at 105 °C until constant weight. All these tests were performed at least in duplicate.

### 2.3. Sample Preparation for Elemental Analysis Offlours

For the determination of elemental differences between the different types of flours, flour samples were prepared as reported by Skendi et al. [23]. Two subsamples of about 6 g from each flour were ashed in a muffle at 650 °C for 5 h. The obtained ash was solubilized with 1 M nitric acid and filtered. The filtrate was made up to the volume of 6 mL with 1 M nitric acid. When needed, dilution was performed with 1 M nitric acid solution. Method blank solution was prepared without using a sample and by applying the same procedure. In general, for all the metals analysis, all glassware and plastic containers used were washed properly, first with nitric acid, then with ultra-pure water, in order to ensure that any contamination does not occur.

### 2.4. Determination of Heavy Metals and Trace and Macro Elements in Flours

The obtained nitric solutions from each flour sample were analyzed for the content of heavy metals, trace elements and macro elements. Samples were analyzed utilizing the method developed by Skendi et al. [24], confirmed for flour matrix. Samples were analyzed in triplicate on the ICP-OES (Inductively Coupled Plasma Optical Emission Spectrometer) model Perkin–Elmer 8300 DV (Perkin-Elmer, Waltham, MA, USA).

The following operating conditions were used: nebulizer flow 0.8 L/min, auxiliary gas flow 15 L/min, sample uptake rate 1.50 mL/min, plasma power 1300 W, and integration time 15 s. Measurements were performed in axial view using the following wavelength (numbers in brackets) in nm for each element: As (188.979), Pb (220.353), Cd (228.802), Zn (206.200), Ni (231.604), Cr (267.716), Fe (238.204), Mn (257.610), Cu (327.393), K (766.490), Na (589.592), Ca (317.933), and Mg (285.213). Calibration was performed using stock solutions of the metal ions appropriately prepared from the purchased standard solutions with ultrapure water. The calibration blank consisted of ultrapure water acidified with nitric acid. Each sample was tested in duplicate, and a blank was run after running ten samples in order to detect possible interferences/contamination. The detection limit was 4.2, 1.4, 2.7, 0.9, 4.6, 9.7, 2.6, 2.8, 5.9, 63.1, 15.0, 79.3, and 10.0 μg/L for As, Cd, Pb, Cr, Cu, Fe, Mn, Ni, Zn, K, Na, Ca, and Mg, respectively.

### 2.5. Sample Preparation for Total Phenolic and Flavonoid Content and Antioxidant Activity

About 0.5 g of wheat flour was double extracted with 2 mL of 70% aqueous methanol in an ultrasonic bath for 15 min taking care that the temperature does not surpass 35 °C. The two supernatants were mixed and centrifuged at 15,339× *g* for 10 min, then stored in the freezer until analysis.

### 2.6. Determination of Total Phenolic and Total Flavonoid Content and Antioxidant Activity in Flours

Determination of total phenolic content (TPC), total flavonoid content (TFC) and antioxidant activity in flours was performed using the same methodology as reported by Skendi et al. [25].

TPC in flour samples was determined using a modified Folin–Ciocalteu method utilizing gallic acid as standard. The absorbance was measured at 725 nm, and TPC was expressed as milligrams of gallic acid equivalents (GAE) per 100 g of dry sample (mg GAE/100 g). On the other hand, TFC was determined using the aluminum complexation method that involves the AlCl_3_ reagent. The absorbance was measured at 510 nm and the results were expressed as milligrams of catechin equivalents per 100 g of sample on a dry weight basis (mg CATE/100 g). Both determinations were performed at room temperature and allowed to rest in the dark.

Antioxidant activity was determined utilizing three different assays: ABTS, DPPH, and FRAP (ferric reducing antioxidant power). In all three assays, was utilized the same Trolox standard and the results were expressed as mg trolox equivalents per 100 g of sample on a dry weight basis (mg TE/100 g). The ABTS antioxidant activity was determined by measuring the absorbance at 734 nm of the final mixture obtained after 4 min of reaction (at room temperature) of the flour extract with the ABTS reagent that has an initial absorbance of 0.70 ± 0.02. The DPPH radical scavenging activity assay was performed allowing the DPPH reagent to react at room temperature with the flour extract for 5 min and recording absorbance at 516 nm. On the other hand, the FRAP assay was carried out by permitting the FRAP reagent to react with flour extract for 4 min at 37 °C. The absorbance was read at 593 nm against a blank. FRAP solution was a mixture of three solutions at proportion of 1:1:10: (a) 20 mM ferric chloride solution, (b) 10 mM TPTZ solution in 40 mM HCl, and (c) 0.3 mM (pH 3.6) acetate buffer, respectively. In general, each determination was completed at least in duplicate.

### 2.7. Sample Preparation for HPLC Analysis

Phenolics were extracted and determined according to the method developed by Skendi, Irakli, Chatzopoulou, and Papageorgiou [25] with some modifications. About 0.25 g of flour was extracted by sonication with 4 mL of 70% aqueous methanol. The obtained extract was first centrifuged, then 2 mL were evaporated under a gentle stream of nitrogen. The residue was reconstituted in 200 µL of mobile phase, filtered through a 0.22 µm nylon membrane filter, and analyzed by HPLC.

### 2.8. HPLC Analysis of Phenolic Acids in Flour

For the measurements, an Agilent LC series 1200 HPLC system was used (Agilent Technologies, Urdorf, Switzerland) equipped with a quaternary gradient pump, a membrane degasser, a Rheodyne injection valve with a 20 µL loop, and a diode array (DAD) and fluorescence (FLD) detectors. Separation of phenolic compounds was conducted on a Nucleosil 100 C_18_ (250 mm × 4.6 mm, 5 µm) column. The separation and determination of phenolics on HPLC were performed by adopting the protocol of Skendi et al. [26]. The extraction procedure and determination were performed in duplicate.

### 2.9. Dough Rheological Properties

Flours were first mixed well into the mixing bowl (300 g) of the Brabender farinograph (Brabender, Duisburg, Germany) that was connected with a circulating water pump and a thermostat which operated at a constant temperature (30 ± 0.2 °C) and tested according to the ICC-standard method 115/1 [27]. The following parameters were obtained from the farinograms: farinograph water absorption (WA), dough development time (DT), and dough stability (ST).

Extensograph tests were performed in Brabender Extensograph (Brabender, Duisburg, Germany) as described in the ICC-Standard 114/1 method [28]. First doughs were prepared in the 300 g mixing bowl of the abovementioned Farinograph with salt and water to produce the dough sample with a consistency of 500 BU (Brabender Units) followed 5 min of mixing. A piece of dough of about 150 g was first rounded into a ball, shaped into a cylinder, and clamped into the holder and allowed to proof for 45, 90, and 135 min in the Extensograph fermenting cabinet (at 30–32 °C). After each proofing time dough was stretched by the Extensograph hook until ruptured. Parameters, such as the resistance to constant deformation after 50 mm stretching (R50 (BU)), maximum resistance to extension (BU), the extensibility (E (mm)) and the Extensional area (A (cm^2^)), were obtained from the Extensograph curve.

### 2.10. Breadmaking and Bread Quality

The bread recipe consisted ofmixing wheat flour (300 g) with salt (6 g), dry yeast (5 g), and water as needed to reach a dough consistency of 500 BU at peak development time. After mixing the ingredients for 5 min in a farinograph bowl there was performed a two-step bulk fermentation (at 30–32 °C) and proofing up to the optimum dough volume. The baking process was 210 °C for at least 23 min for a dough piece of 100 g. The bread loaves were cooled at room temperature before measuring their weight and volume by the rapeseed displacement method. The specific volume of each bread was obtained from loaf volume divided by weight. Cooled bread loaves were sealed in polyethylene bags and stored in the freezer (at 7 °C) for the time needed to perform other quality measurement tests. At least nine loaves for each flour recipe were baked.

Moisture and texture analysis of bread crumbs were performed after 1, 4, and 8 days of storage as described previously by Koletta et al. [29]. In order to test the rheological properties of bread, the firmness and moisture of bread were measured after 1, 4, and 8 days of storage at 7 °C, using a texture analyzer, model TA-XTplus (Stable Micro Systems, Godalming, Surrey, UK). Two slices of about 25 mm thickness were obtained from the middle of each loaf. After the crust removal, each bread crumb was subjected to a compression test according to the AACC method 74–09 [30]. Bread firmness for each recipe and sampling time was calculated based on the results obtained from the measurements of at least three bread loaves.

### 2.11. Statistical Analysis

The determination of the effect of different flours on the dough and bread quality was carried out in two replications. One-way analysis of variance (ANOVA) followed by Duncan’s multiple range test was used to determine significant differences between the means of dough and bread quality parameters. A *t*-test was used to detect significant differences among the group of refined flours and whole meal wheat flours. Pearson’s correlation analysis was used to assess the possible relationship among the parameters studied. Statistical analyses were performed using IBM SPSS Statistics for Windows (Version 25.0, IBM Corp. Armonk, NY, USA) taking into account a significant level of ≤0.05.

## 3. Results and Discussion

### 3.1. Quality of Flours

Flour quality characteristics are shown in Table 1. In the studied wheat flours, the moisture varied between 10.51% and 13.79%, and the ash content from 0.65% to 2.38%. Whole meal flours showed higher moisture levels (11.07–13.79%) than refined ones (10.51–10.96%). It is clear that ash content depends on the extraction rate, the higher the extraction rate, the lower the ash content. Among the flour samples, the einkorn landrace showed the highest ash content, followed by durum, and finally, soft wheat (landrace and commercial cultivar). Landraces of the present study showed higher ash content than those observed by Visioli et al. [31] for Sicilian flours obtained from wheat landraces (1.6–2.0%). Our results are similar to the levels reported by Geisslitz et al. [32] for whole meal flours from commercial wheat varieties cultivated in Germany. The variation in the ash content is related not only to the soil geochemical composition and agronomic practices adopted, but also to the milling process applied and the flour extraction rate.

The resistance that the falling plunger finds in the flour-water mixtures, that is falling number (FN), varied between 372.5 s and 793.5 s for M90 (einkorn) and DuB70 (durum), respectively. On the other hand, commercial emmer samples showed values of 486.0 s and 480.5 s for whole meal flour and 70% extraction rate, respectively. Only the soft wheat S100 sample exhibited similar FN value. In general, FN values around 300 s are linked with a minimal amylase activity and high end-use quality, but practically this threshold changes depending on each country’s market requirements [33]. In addition, in their review, He, Lin, Chen, Tsai, and Lin [33], stressed that factors, such as cultivar, environment, and agronomic practices, affect starch paste viscosity and consequently the FN values. Very high values of FN observed for landrace flours could be explained by low a-amylase activity, but also differences in starch/protein quality, and their ability to increase the viscosity in the water mixtures and delay the time of falling plunger should be considered. In those cases, the flours should be supplemented with a form of amylolytic enzyme or with malted grain flours.

Total protein content varied from 9.98% to 16.62%, with the einkorn sample showing the highest level and the soft wheat flour showing the lowest. Similarly, in the study by Tran, Konvalina, Capouchova, Janovska, Lacko-Bartosova, Kopecky, and Tran [12] common wheat and bread wheat landraces had the lowest protein content among einkorn and emmer wheat species. Moreover, Longin, Ziegler, Schweiggert, Koehler, Carle, and Würschum [1] showed that in spite of the limited nitrogen supply, einkorn, emmer, and spelt had higher protein content than bread wheat. Differences in the protein content exist due to genetic factors and environmental factors. Factors, such as cultivation area, harvesting year, fertilization, and milling, are also affecting the final values of protein content. In their study, Geisslitz, Wieser, Scherf, and Koehler [32] found higher protein levels (13.0–15.5%) in commercial durum wheat cultivars than in the present study. No significant differences were observed in the protein content among emmer samples of different extraction rate or origin (commercial or landrace), with the exception of D70 that was found to have significant higher protein content (14.26%).

Gluten represents the most important constituent of wheat for baked products due to the strength and texture it provides to the baked wheat products. For a baker, a higher gluten content is linked with higher quality bread. It was not possible to determine wet/dry gluten content for einkorn because the gluten present was not able to form a network, and therefore, the mixture was washed (passed) through the filter. The wet gluten yield of samples ranged between 21.54% (emmer DC100) and 38.33% (durum DuA70) whereas the dry gluten ranged from 7.82% (durum DuB100) to 13.2% (durum DuA70) (Table 1). In Europe, wheat flours with wet gluten of more than 27% are considered suitable for breadmaking applications [34]. The above threshold is not met for four samples of Table 1, namely, M90, D100, DuB100, and DC100.Significant statistical differences were observed among the samples. Despite the relatively low protein content of durum wheat samples they exhibited high values of wet gluten with DuA70 showing the highest wet gluten and dry gluten contents (38.33% and 13.20%, respectively).Tran, Konvalina, Capouchova, Janovska, Lacko-Bartosova, Kopecky, and Tran [12] reported common wheat with comparatively lower wet gluten content (30.87%) to bread wheat landraces (33.07%), emmer wheat (37.96%), and einkorn (36.35%) samples. In their study, Kulkarni et al. [35] reported wet gluten values varying from 20.9% to 42.1% whereas dry gluten from 9.4% to 24.5% in hard red winter and hard red spring wheat flours. Moreover, in their study, they report that the gluten amount in flour is depended on the degree and type of milling. In general, refined flours studied showed lower ash and moisture contents, and higher wet and dry gluten than their whole meal wheat counterparts.

### 3.2. Microelements in Flours

Mineral micronutrient intake is considered important for humans’ health and depending on their needed amount they are categorized into two groups: macro-minerals (Ca, Mg, K, and P) and micro-minerals (Cu, Zn, Fe, and Mn). The amount of minerals in flours varies depending on a great number of factors, such as genetics, climate conditions, soil type, fertilization, etc. In the literature, the amount of minerals in wheat flour varies among the studies. It was reported that organically grown wheat genotypes could be used to increase mineral intake [36].

Flours reported in this study vary in their content of micronutrients (Figure 1): minor microelements Fe (12.2–67.59 mg/kg), Cu (2.45–20.19 mg/kg), Zn (9.05–61.73 mg/kg), and Mn (8.86–35.55 mg/kg); major microelements P (488.87–2354.54 mg/kg), K (2590.46–4937.88 mg/kg), Ca (85.42–728.38 mg/kg), and Mg (370.62–1896.75 mg/kg). In general, the more refined the flour the lower the amount of metals present. Moreover, the landraces showed a higher level of microelements than the respective commercial samples. In the study of Rodríguez et al. [37], wheat landraces from the Canary Islands showed that minor microelements Fe (32.6–46.6 mg/kg), Cu (0.70–4.07 mg/kg), Zn (26.6–36.5 mg/kg), and Mn (15.1–27.5 mg/kg); and major microelements P (1948–2829 mg/kg), K (3663–4985 mg/kg), Ca (288–481 mg/kg), and Mg (885–1299 mg/kg) vary largely among the 175 samples studied. The wheat Greek landraces of the present study are higher in minor microelements than the organic commercial wheat products sold in the Greek market [23]. Nevertheless, besides landrace, factors, such as milling procedure, cultivation year, and climatic and soil conditions, can be the reason behind the differences observed. Extraction rate significantly affected the level of the Fe, Cu, Zn, Ni, Ca, and Mg in the present study with whole meal wheat flours showing higher levels than refined flours.

In the present study, the levels of heavy metals vary among the samples: Cd (0.98–20.89 µg/kg), Pb (0.00–17.22 µg/kg), Ni (30.93–429.08 µg/kg), Cr (8.44–86.38 µg/kg), and As (0.00–18.43 µg/kg). All the samples showed levels of Cd and Pb much lower than the limits set by European Commission Regulation (EC) No 1881/2006 for cereals.

Some authors report that domesticated emmer wheat accessions have higher Zn levels than modern durum and bread durum genotypes [38]. In their study Zhao et al. [39] monitored the variation in mineral micronutrient concentration in 175 wheat lines of diverse origins. They reported a great variation in Fe (28.9–50.8 mg/kg) and Zn (13.5–34.5 mg/kg) content. Among the wheat lines that they have studied, einkorn showed the highest Fe levels followed by spelt, emmer, bread wheat, and durum wheat. On the other hand, they observed no significant difference in the levels of Zn.

### 3.3. Phenolics and Antioxidant Activity of Flours

Total phenolic content (TPC), total flavonoid content (TFC), and antioxidant activity of the flours are shown in Figure 2. The values of TPC varied from 354.60 µg/kg to 607.28 µg/kg, whereas those of TFC ranged from 0.38 µg/kg to 2.79 µg/kg. In general, all the landraces showed higher TPC values than the commercial emmer samples, except S70. On the other hand, the samples M90, D100, DuA100, and DuB100 performed best on the TFC, all of them exhibiting values higher than 2.2 µg CATE/g. Not surprisingly, they are all whole meal flours except the einkorn M90 which has still a high extraction rate (90%). As it concerns TPC, the best performers were again M90, D100, and DuA100, with the addition of S100 and D70. All the above belong to a group with TPC higher than 450 µg GAE/g of flour. D70 outreached its commercial counterpart DC70 and the same stands for D100 when compared to DC100.

Antioxidant activity measured with ABTS, DPPH, and FRAP varied significantly, with einkorn-M90 showing the highest values among the flours studied in all three tests. Contrary Serpen et al. [40], reported a higher total antioxidant activity in emmer wheat samples than in einkorn. According to our results, the emmer landrace (whole meal-D100) showed the second higher antioxidant activity when measured with DPPH and ABTS and performed best along M90 in FRAP; D70 was the next in the range. Both emmer landrace flours exhibited higher antioxidant activity than their commercial counterparts. The refined flours showed significantly lower antioxidant activity compared to the whole meal wheat flours.

Having a closer look at the profile of phenolic acids present in the flours (Table 2) one can observe that except for the sinapic acid that was not detected in the commercial flours and found only in trace amount in one durum landrace (DuA70), in all the other samples there were detected, also protocatechuic acid, 4-hydroxybenzoic acid, vanillic acid, p-coumaric acid, and ferulic acid. Among the phenolic acids present, ferulic acid prevailed in all the wheat landraces and commercial samples with the highest content observed for the einkorn sample-M90 (10.45 µg/g of dry basis flour). The amount of ferulic acid in this study (1.44–10.45 µg/g) was higher than that observed by Li, Shewry, and Ward [7] (1.2–6.2 µg/g). Whole meal emmer samples exhibited highest content in protocatechuic acid (D100),4-hydroxybenzoic acid (DC100, followed by D100), and vanillic acid (DC100, followed by D100), whereas D70 exhibited the highest content of sinapic acid. Regarding p-coumaric acid samples M90, D100, and DuA100 were those with the highest content observed in this study.

The 90% extraction flour of the einkorn exhibited the highest content of phenolic acids (19.14 µg/g of flour) whereas the commercial refined emmer flour had the lowest (5.92 µg/g of flour). Einkorn wheat was found by Li, Shewry, and Ward [7] in their study among 175 samples of wheat flour with the highest content of total phenolic acids. The milling process affected the amount of phenolic compounds studied (except for S70) with refined flours showing significantly lower values than the whole meal wheat flours.

In their study Laddomada et al. [41] reported for five whole meal flour of Italian durum wheat cultivars much lower values of free phenolic acids than those of the two Greek durum landraces. The landrace wheat flours of the present study were higher in free ferulic acid (1.44–10.45 vs. 1.31–1.79 µg/g), p-coumaric acid (1.23–1.66 vs. 0.44–0.82 µg/g), vanillic acid (0.83–2.99 vs. 1.20–1.33 µg/g), sinapic acid (0.36–1.30 vs. <0.01), and 4-hydroxybenzoicacid (0.93–2.69 vs. 0.28–0.77 µg/g) than in twelve hard red winter wheat varieties grown in Kansas, USA with exception of one sample that was higher in p-coumaric acid (5.62 µg/g) [42].

### 3.4. Dough Rheology

The rheological parameters of the doughs measured with the farinograph and extensiograph are shown in Figure 3. Water absorption capacity shows the ability of the dough to hold water during processing with flours having higher protein content showing higher water absorption [34]. The water absorption varies from 53.0% to 71.9% with the two durum wheat fours showing the highest values (Figure 3a). Whole meal wheat flours showed significantly higher water absorption capacity than refined flours, this was due to the higher ability of fibers to absorb water [43].

Dough development time (DDT) is influenced by the quantity and the quality of protein and the amount of starch, its size and level of degradation. According to Tran, Konvalina, Capouchova, Janovska, Lacko-Bartosova, Kopecky, and Tran [12], wheat flours show values that vary from 0.99 min to 7.36 min with flours with longer time indicating better breadmaking quality. The above study reported the longest DDT for bread wheat than einkorn and emmer wheat species. This is valid for whole meal landrace flours; durum and soft wheat flours showed significantly higher DDT values than emmer and einkorn landrace. Commercial emmer whole meal flour showed similar DDT with that of whole meal soft wheat landrace. No significant difference was observed between the DDT values of refined flours.

Einkorn showed the smaller dough stability (DS) and highest degree of softening (SD) among the studied samples, revealing low resistance during mixing, which means poor technological quality of this flour. The highest DS value was observed in one durum landrace (DuB100) whereas the lowest SD value was observed in refined commercial emmer flour (DC70). The higher degree of softening and the lower stability are characteristics of a weaker gluten network [34]. The strength of the gluten network can derive from the amount and the quality of the gluten but also the amount and the type of bran (fibers) present in the flour.

The variation in the energy (E), Resistance at 50 mm stretching (R50) and Extensibility (Ex) at 45 min and 135 min of proofing is shown in (Figure 3b). The einkorn dough was not possible to be measured in the extensiograph because of its loose dough structure after 45 min of proofing. The values of extensiograph parameters varied largely among the samples after 45 min (E (12.5–127.5 cm^2^), Ex (59–187.5 mm) and R50 (51.5–396.0 BU)) and 135 min (E (12.5–116.5 cm^2^), Ex (51.5–164.5 mm) and R50 (70.0–479.0 BU)) of proofing. Among the flours studied, one of the refined durum wheat landraces (DuA70) showed the lowest value for the parameters E, Ex, and R50 at 45 min of resting. Refined flours showed significantly higher extensibility values than the whole meal flours at 45 min and 135 min of proofing. Commercial whole meal emmer showed similar extensibility with whole meal emmer (D100) at 45 min of proofing, but is higher energy and resistance to extension. On the other hand, at 135 min of proofing there were observed no significant differences between the two samples. Regarding the refined emmer flours, commercial flour showed significantly higher values of energy, extensibility, and resistance to extension at 45 min and 135 min of proofing compared to emmer landrace except for extensibility at 135 min where there was no difference observed.

According to the literature [1,44], flours of ancient wheat species yield softer doughs with low elasticity and high extensibility compared to common wheat because of the poor gluten quality. This means that a different approach must be applied to ancient flours in order to obtain the desired bread quality.

### 3.5. Breadmaking Quality of Flours

The moisture and firmness of breads at 1, 3 and 8 days of storage at 7 °C and the specific volume of the breads is shown in Table 3. The specific volume of the bread depends on the dough’s capacity to retain the gas formed during fermentation. It was observed a significant decrease in the specific volume of the refined flours in the following order: commercial emmer > soft wheat landrace, durum wheat landrace flours (DuA70 > DuB70), and emmer landrace. Regarding whole meal flours, the specific volume of the breads decreases in the following order: einkorn landrace > soft wheat landrace = commercial emmer > emmer landrace = durum landrace (DuA100) > durum landrace (DuB100). In their study, [1] recommend techniques used for “traditional” dough production in the bakery utilizing reduced energy input and increasing resting times during dough production in order to increase loaf volume of einkorn, emmer, and spelt breads.

Durum landrace DuB100 showed the lowest specific bread volume. This bread showed the highest moisture content among the samples on the first day of storage whereas refined soft wheat landrace showed the lowest moisture content. Only samples M90, DuA100, DuB100, and DuA70 showed decreased moisture on the fourth day of storage. Among the rest of the samples that kept the moisture content stable for four days, only soft wheat landrace (whole meal and refined flour) did not show any significant change in the moisture content during the eight days of storage.

During the first day of storage, samples D100 and DuB100 showed the highest firmness values among the samples whereas the DC70 was the lowest. Moisture and specific volume can affect crumb texture increasing its hardness if moisture and specific volume of the bread is decreased. Thus, it is expected that the firmness of the bread increases with storage time since the moisture of the bread is decreased. It was observed that although the moisture content did not change with time, the firmness of the samples was increased. Correlation among the three parameters (specific volume of the bread, moisture, and firmness) showed that the firmness of the bread crumb was negatively correlated with the specific volume (correlation coefficient 0.846, 0.803, and 0.752 for firmness at days 1, 4, and 8, respectively, with correlation significance at the 0.01 level), but not with the moisture content of the bread.

## 4. Conclusions

The objective of this manuscript was to evaluate wheat flours from Greek landraces regarding their chemical composition, elemental analysis, phenolic acid profile and content, antioxidant activity, dough rheology, and breadmaking quality. Wide variation was identified for all the parameters tested. The differences were not only in the dough rheology and bread quality, but also from a nutritional point of view were observed among the different wheat landraces and between the conventionally grown commercialemmers-DC100 and D70, and the landrace counterparts D100 and D70 that was organically grown. The studied landraces were higher in microelements, phenolics, and antioxidant activity compared to commercial emmer millings. In the present study, we verified that whole grain breads have lower volume and harder crumbs than breads from refined wheat flour. Despite the dough rheological profile reported in this study for certain samples, using appropriate breadmaking procedures could lead to the production of breads with acceptable quality in addition to a denser nutritional profile when the examined wheat landraces are used. Moreover, the landraces studied could serve as a genetic pool for the breeders to improve wheat nutritional quality.

## Figures and Tables

**Figure 1 foods-12-01618-f001:**
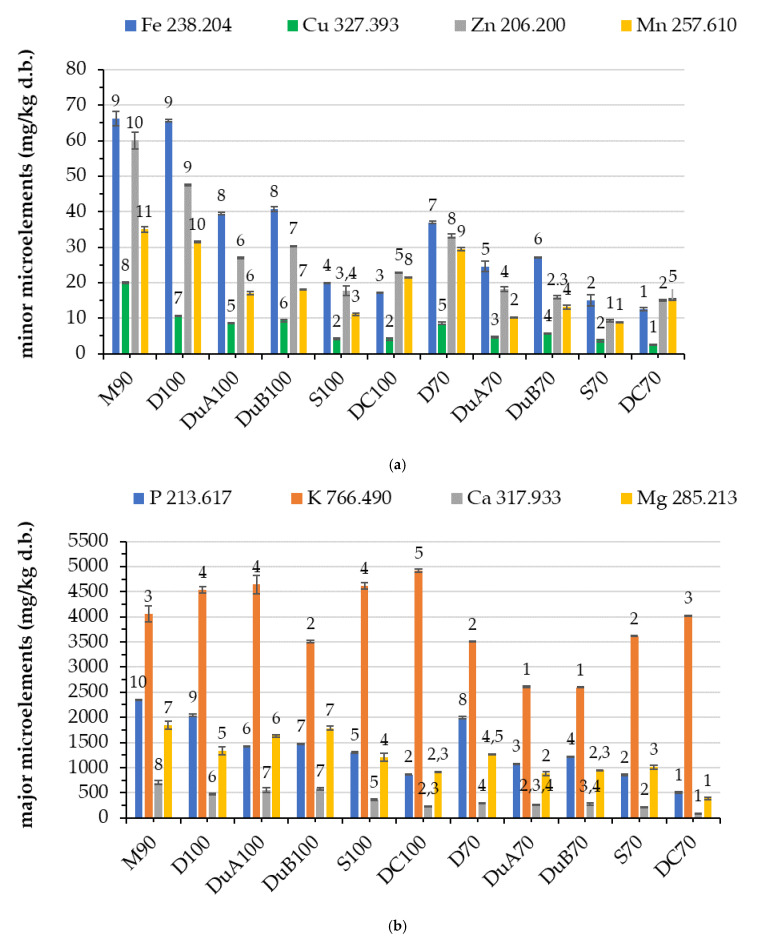
Concentration of micronutrients in wheat flour samples: (**a**) Minor micronutrients (mg/kg dry basis); (**b**) Macronutrients (mg/kg dry basis); and (**c**) Heavy metals (mg/kg dry basis). Results are reported as mean values ± standard deviation (bars). Different superscripts numbers above the error bars for each reported parameter indicate significant differences (*p* ≤ 0.05) among the means, as determined by the Duncan’s multiple range test.

**Figure 2 foods-12-01618-f002:**
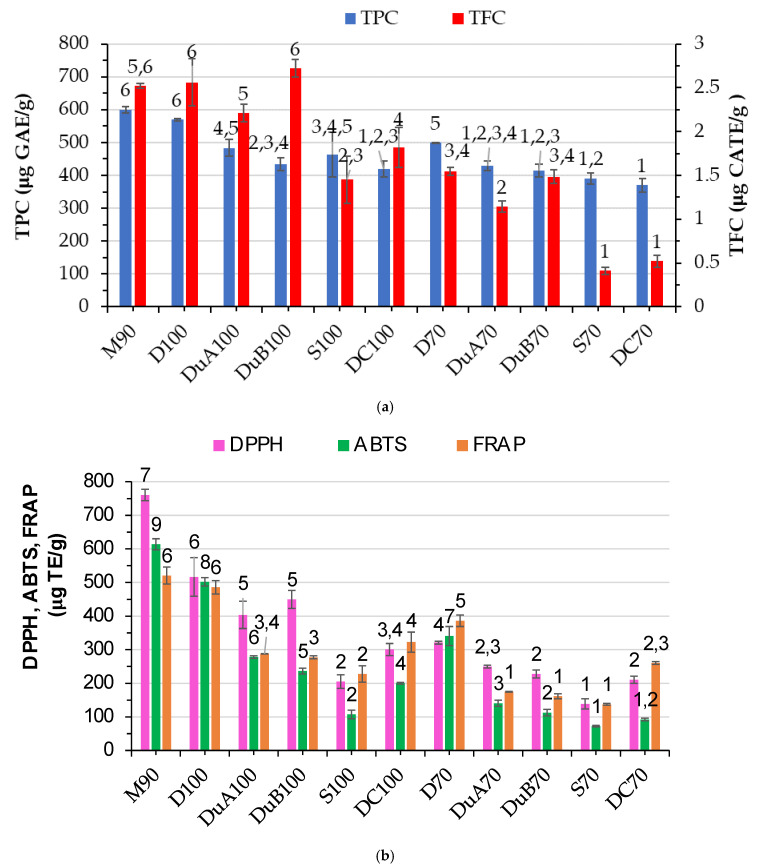
Total phenolics content (TPC), total flavonoids content (TFC) (**a**) and antioxidant activity as evaluated by DPPH, ABTS, and FRAP tests (**b**) in wheat flours. Results are reported as mean values ± standard deviation (bars). Different superscripts numbers above the error bars for each reported parameter indicate significant differences (*p* ≤ 0.05) among the means, as determined by the Duncan’s multiple range test.

**Figure 3 foods-12-01618-f003:**
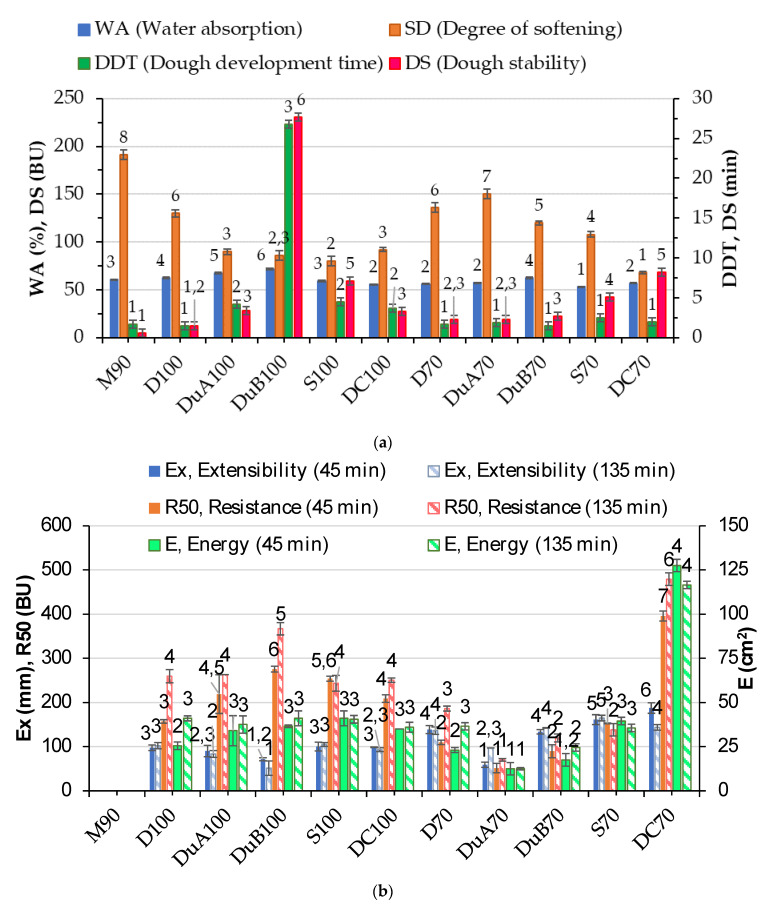
Variation in the farinograph (**a**) and extensograph (**b**) parameters of doughs. Different numbers above error bars of each parameter reported to indicate significant differences (*p* = 0.05) among the means, as determined by the Duncan’s multiple range test. Einkorn dough (M90) could not be measured in extensiograph due to its loose structure.

**Table 1 foods-12-01618-t001:** Chemical analysis of wheat flours on a dry-weight basis ^a^.

Samples	Ash (%)	Moisture (%)	Falling Number (s)	Protein (%)	Wet Gluten (%)	Dry Gluten (%)
M90 *	2.38 ± 0.00 ^9^	11.07 ± 0.02 ^1,2,3^	372.5 ± 0.7 ^1^	16.62 ± 0.11 ^6^	0.00 ± 0.00 ^1^	0.00 ± 0.00 ^1^
D100	2.12 ± 0.01 ^8^	12.91 ± 0.70 ^5^	645.0 ± 1.4 ^7^	13.16 ± 0.13 ^4^	23.28 ± 2.32 ^2^	8.60 ± 0.61 ^2^
DuA100	1.92 ± 0.01 ^7^	11.69 ± 0.28 ^3,4^	571.5 ± 3.5 ^6^	11.05 ± 0.06 ^2^	34.53 ± 0.08 ^4^	12.01 ± 0.09 ^5^
DuB100	1.92 ± 0.01 ^7^	11.58 ± 0.45 ^2,3,4^	518.5 ± 6.4 ^5^	11.34 ± 0.24 ^2^	21.54 ± 0.93 ^2^	7.82 ± 0.37 ^2^
S100	1.88 ± 0.01 ^6^	12.10 ± 0.47 ^4^	407.5 ± 0.7 ^2^	9.98 ± 0.08 ^1^	28.27 ± 1.87 ^3^	11.00 ± 0.43 ^3,4^
DC100	1.89 ± 0.00 ^6^	13.79 ± 0.12 ^6^	486.0 ± 2.8 ^4^	13.41 ± 0.28 ^4^	20.33 ± 1.63 ^2^	8.10 ± 0.36 ^2^
D70	1.47 ± 0.03 ^5^	10.54 ± 0.46 ^1^	738.5 ± 3.5 ^9^	14.26 ± 0.22 ^5^	33.99 ± 1.25 ^4^	11.12 ± 0.22 ^3^
DuA70	1.19 ± 0.01^3^	10.96 ± 0.05 ^1,2,3^	663.5 ± 0.7 ^8^	11.86 ± 0.16 ^3^	38.33 ± 2.05 ^5^	13.20 ± 0.35 ^6^
DuB70	1.26 ± 0.00 ^4^	10.51 ± 0.33 ^1^	793.5 ± 3.5 ^10^	11.81 ± 0.05 ^3^	34.52 ± 0.62 ^4^	11.54 ± 0.58 ^4,5^
S70	0.78 ± 0.01 ^2^	10.79 ± 0.12 ^1,2^	465.5 ± 4.9 ^3^	10.21 ± 0.10 ^1^	28.60 ± 1.00 ^3^	10.28 ± 0.21 ^3^
DC70	0.65 ± 0.00 ^1^	10.56 ± 0.11 ^1^	480.5 ± 3.5 ^4^	13.50 ± 0.09 ^4^	28.66 ± 0.14 ^3^	12.24 ± 0.15 ^5^

^a^ Values are means and standard deviation of at least two determinations. Different superscripts numbers in the same column indicate differences (*p* ≤ 0.05), amongst the means, as determined by the Duncan’s multiple range test. * The mixture formed was completely washed out the filter.

**Table 2 foods-12-01618-t002:** Content (µg/g) of phenolic compounds on a dry-weight basis of wheat flours ^a^.

Samples	Phenolic Acids (µg/g)						
	Protocatechuic Acid	4-hydroxybenzoic Acid	Vanillic Acid	p-Coumaric Acid	Ferulic Acid	Sinapic Acid	Totalphenolic Acids
M90	1.86 ± 0.02 ^7^	1.78 ± 0.03 ^8^	2.24 ± 0.02 ^7^	1.63 ± 0.01 ^6^	10.45 ± 0.13 ^8^	1.18 ± 0.02 ^8^	19.14
D100	1.95 ± 0.02 ^8^	2.23 ± 0.03 ^9^	2.67 ± 0.02 ^9^	1.66 ± 0.02 ^6^	7.46 ± 0.11 ^7^	0.92 ± 0.01 ^6^	16.90
DuA100	1.42 ± 0.02 ^6^	1.39 ± 0.01 ^5^	2.58 ± 0.01 ^8^	1.65 ± 0.02 ^6^	4.25 ± 0.10 ^5^	0.45 ± 0.02 ^3^	11.74
DuB100	1.45 ± 0.02 ^6^	1.79 ± 0.02 ^8^	1.32 ± 0.01 ^3^	1.57 ± 0.02 ^5^	7.32 ± 0.02 ^7^	1.03 ± 0.02 ^7^	14.48
S100	1.00 ± 0.02 ^5^	1.63 ± 0.03 ^7^	1.47 ± 0.02 ^4^	1.23 ± 0.02 ^1^	6.12 ± 0.91 ^6^	0.36 ± 0.02 ^2^	11.81
DC100	0.96 ± 0.02 ^4,5^	2.69 ± 0.02 ^10^	2.99 ± 0.02 ^10^	1.36 ± 0.02 ^3^	3.61 ± 0.03 ^4,5^	Nd ± 0.00 ^1^	11.61
D70	0.80 ± 0.03 ^2^	1.27 ± 0.02 ^4^	1.53 ± 0.02 ^5^	1.30 ± 0.01 ^2^	2.33 ± 0.04 ^2^	1.30 ± 0.02 ^9^	8.53
DuA70	0.85 ± 0.02 ^3^	1.12 ± 0.02 ^3^	1.51 ± 0.02 ^5^	1.47 ± 0.01 ^4^	3.08 ± 0.42 ^3,4^	LLQ ^1^	8.03
DuB70	0.75 ± 0.01 ^1^	1.47 ± 0.02 ^6^	1.65 ± 0.02 ^6^	1.24 ± 0.02 ^1^	2.52 ± 0.02 ^2,3^	0.68 ± 0.02 ^4^	8.31
S70	0.95 ± 0.02 ^4^	0.93 ± 0.02 ^1^	0.97 ± 0.01 ^2^	1.30 ± 0.02 ^2^	1.44 ± 0.02 ^1^	0.85 ± 0.02 ^5^	6.44
DC70	0.84 ± 0.01^2,3^	1.02 ± 0.02 ^2^	0.83 ± 0.02 ^1^	1.33 ± 0.02 ^2,3^	1.90 ± 0.02 ^1,2^	Nd ^1^	5.92

LLQ, lower than LOQ (limit of quantification), ND, not detected. ^a^ Values are means of duplicate analysis. Different superscripts numbers in the same column indicate differences (*p* ≤ 0.05) amongst the means, as determined by the Duncan’s multiple range test.

**Table 3 foods-12-01618-t003:** Variation in specific volume, moisture and firmness of the breads made from wheat flours *.

Sample	Specific Volume (mL/g)	Moisture (%)			Firmness (N)		
		Day 1	Day 4	Day 8	Day 1	Day 4	Day 8
M90	1.9 ± 0.1 ^5^	39.6 ± 0.8 ^2,3,4,c^	36.5 ± 0.9 ^1,2,3,b^	30.9 ± 0.4 ^1,a^	33.0 ± 4.5 ^4,a^	59.0 ± 5.9 ^6,b^	103.7 ± 10.5 ^4,5,c^
D100	1.6 ± 0.0 ^2^	40.9 ± 0.6 ^3,4,b^	40.3 ± 0.7 ^3,b^	34.0 ± 1.1 ^2,3,4,a^	59.1 ± 2.3 ^6,a^	71.0 ± 2.8 ^7,b^	106.2 ± 10.4 ^5,c^
DuA100	1.5 ± 0.1 ^2^	41.2 ± 0.4 ^4,c^	36.9 ± 0.1 ^1,2,3,b^	35.1 ± 0.9 ^3,4,a^	43.6 ± 4.3 ^5,a^	58.6 ± 3.5 ^6,b^	83.9 ± 7.8 ^3,c^
DuB100	1.3 ± 0.0 ^1^	45.1 ± 0.8 ^5,b^	39.9 ± 1.0 ^2,3,a^	39.9 ± 0.1 ^5,a^	60.7 ± 4.4 ^6,a^	79.3 ± 3.5 ^8,b^	96.1 ± 4.7 ^4,c^
S100	1.7 ± 0.0 ^3^	40.1 ± 0.6 ^3,4,a^	38.6 ± 2.9 ^1,2,3,a^	34.8 ± 2.5 ^3,4,a^	26.9 ± 2.6 ^3,a^	53.1 ± 5.6 ^5,b^	80.6 ± 3.4 ^3,c^
DC100	1.7 ± 0.1 ^3^	37.6 ± 1.8 ^1,2,b^	34.8 ± 0.5 ^1,a,b^	32.5 ± 0.5 ^1,2,3,a^	44.9 ± 2.3 ^5,a^	95.5 ± 6.3 ^9,b^	133.4 ± 7.6 ^6,c^
D70	1.8 ± 0.0 ^4^	38.7 ± 1.4 ^1,2,3,b^	36.5 ± 1.4 ^1,2,3,b^	31.8 ± 1.5 ^1,2,a^	31.8 ± 2.0 ^4,a^	44.2 ± 3.8 ^4,b^	80.9 ± 6.3 ^3,c^
DuA70	2.1 ± 0.1 ^6^	40.8 ± 0.2 ^3,4,b^	35.2 ± 2.3 ^1,a^	33.4 ± 0.3 ^1,2,3,a^	20.1 ± 2.7 ^2,a^	29.0 ± 3.2 ^2,b^	51.4 ± 9.7 ^2,c^
DuB70	1.9 ± 0.1 ^5^	41.4 ± 1.0 ^4,b^	38.8 ± 1.5 ^1,2,3,a,b^	36.6 ± 1.4 ^4,a^	20.1 ± 2.6 ^2,a^	35.4 ± 2.9 ^3,b^	52.2 ± 7.9 ^2,c^
S70	2.4 ± 0.1 ^7^	36.7 ± 1.4 ^1,a^	36.0 ± 3.3 ^1,2,a^	31.6 ± 1.8 ^1,2,a^	17.3 ± 1.5 ^2,a^	38.5 ± 6.0 ^3,b^	56.7 ± 5.3 ^2,c^
DC70	2.9 ± 0.0 ^8^	41.0 ± 0.2 ^3,4,b^	40.1 ± 0.3 ^2,3,b^	36.6 ± 0.8 ^4,a^	7.2 ± 0.6 ^1,a^	12.3 ± 1.6 ^1,b^	22.6 ± 1.9 ^1,c^

* Values are means and standard deviation of at least six determinations. Different superscripts numbers in the same column indicate differences (*p* ≤ 0.05) amongst the means, as determined by the Duncan’s multiple range test. Different superscripts letters in the same row within the same parameter measured indicate differences (*p* ≤ 0.05) amongst the means, as determined by the Duncan’s multiple range test.

## Data Availability

Data is contained within the article.

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
