# Peer review of "Greek Landrace Flours Characteristics and Quality of Dough and Bread"

_foods, 2023, doi:10.3390/foods12081618_

Round 1

Reviewer 1 Report

Abstract: Add numeric values in this section.

Materials and methods

Prepare a separate paragraph for Extract preparation. Don't add extract preparation method in the analytical part. 

Elaborate the conditions used during DPPH, ABTS and FRAP assays.

Section 

Quality of flours

Improve the language of this section. 

Flour quality characteristics are shown in Table 1. The moisture in wheat flours var- 254 ied between 10.51 and 13.79% and the ash content was from 0.65 to 2.38%.Whole-meal 255 wheat flours showed higher moisture content (11.07-13.79%) than refined flours (10.51- 256 10.96%). It is clear that ash content depends on the extraction rate, the higher the extrac- 257 tion rate the lower the ash content. Among the flour samples, einkorn landrace showed 258 the highest ash content followed by durum and finally soft wheat (landrace and com- 259 mercial cultivar). Landraces of the present study showed higher ash content than those 260 observed by Visioli, et al. [24] for flours obtained from wheat landraces of Sicily, Italy 261 (1.6-2.0%)but similar to the results reported by Geisslitz, et al. [25] for whole-meal wheat 262 flours of commercial varieties cultivated in Germany.

Authors just compared their results with other studies. Reason behind specific observations are not mentioned. 

Reviewer 2 Report

Manuscript: Greek landrace flours characteristics and quality of dough and bread

Manuscript Number: foods-2280650

Comments:

In the abstract: An abstract is often presented separately from the article, so it must be able to stand alone. add some quantitative data

Keywords→ arrange in alphabetical order.

P3, L131, L133 & P5, L240: References ICC 104/1 (ICC Standards 1994), AACC 56-81.04 (AACC 2012) and AACC method 74-09 (2000) are missing in the reference list.

P4, Figure 2 (b): Correct Vertical axis title → ABTS, DPPH, FRAP (μg TF/g)

P4, Figure 2 (b): Correct Vertical axis title → add units for ABTS, DPPH

P4, Figure 2: All of flavonoids are a group of natural substances with variable phenolic structures. You report total flavonoid content (TFC) more than total phenolic content (TPC). How can it possible? Please explain

Why is the difference in the amount of total phenolic content in Figure 2 and Table 2 big? for example, sample D100 (Moisture (%): 12.9): Figure 2: Total phenolic content about 560 (μg GAE/g), Table 2: Total phenolic 16.9 (μg /g)

Why did you choose the DPPH method for antioxidant activity determination?

P4, Table 2: Sample M100?

In this paper, the methodology is well written, but results and discussion need to be improved. Results presented need a better discussion.

Reviewer 3 Report

Detailed recommendation:

Abstract: in my opinion you should add more data to the abstract section.

Key words: please add: gluten

Why were the tests performed only in two replicates? In scientific practice, the tests are performed in triplicate.

In my opinion, for the full characteristics of the flour, the characteristics of gluten (gluten quality) are indicated.

The article lacks information on the appearance of the bread crumb, porosity, external appearance. Has an organoleptic evaluation been carried out? The results could be interesting.

Round 2

Reviewer 1 Report

Now, there is no issues in your manuscript. 

Author Response

Thank you for the appreciation of our work

Reviewer 2 Report

P4, Figure 2 (b): Correct Vertical axis title → ABTS, DPPH, FRAP (μg TE/g) (Remove extra DPPH)

Author Response

Thank you for your comment on Figure 2b. We revised the Figure accordingly.

Reviewer 3 Report

The manuscript has been revised as suggested by the Reviewer.

Author Response

(The authors gave the same response as above.)
